# Tryptophan and Kynurenine Enhances the Stemness and Osteogenic Differentiation of Bone Marrow-Derived Mesenchymal Stromal Cells In Vitro and In Vivo

**DOI:** 10.3390/ma14010208

**Published:** 2021-01-04

**Authors:** Hai Thanh Pham, Mitsuaki Ono, Emilio Satoshi Hara, Ha Thi Thu Nguyen, Anh Tuan Dang, Hang Thuy Do, Taishi Komori, Ikue Tosa, Yuri Hazehara-Kunitomo, Yuya Yoshioka, Yasutaka Oida, Kentaro Akiyama, Takuo Kuboki

**Affiliations:** 1Department of Oral Rehabilitation and Regenerative Medicine, Dentistry and Pharmaceutical Sciences, Okayama University Graduate School of Medicine, Okayama 700-8558, Japan; pth_uytki@yahoo.com (H.T.P.); thuharhm@gmail.com (H.T.T.N.); ppjj5xuj@s.okayama-u.ac.jp (A.T.D.); pa62884x@s.okayama-u.ac.jp (H.T.D.); de19016@s.okayama-u.ac.jp (T.K.); de421035@s.okayama-u.ac.jp (I.T.); de19034@s.okayama-u.ac.jp (Y.H.-K.); de17054@s.okayama-u.ac.jp (Y.Y.); gmd20103@s.okayama-u.ac.jp (Y.O.); akentaro@md.okayama-u.ac.jp (K.A.); kuboki@md.okayama-u.ac.jp (T.K.); 2Faculty of Dentistry, Hai Phong University of Medical and Pharmacy, Haiphong 04211, Vietnam; 3Department of Molecular Biology and Biochemistry, Dentistry and Pharmaceutical Sciences, Okayama University Graduate School of Medicine, Okayama 700-8558, Japan; 4Department of Biomaterials, Dentistry and Pharmaceutical Sciences, Okayama University Graduate School of Medicine, Okayama 700-8558, Japan

**Keywords:** amino acid, mesenchymal stromal cells, stemness, tryptophan, kynurenine, osteogenesis, adipogenesis, screening, injury/fracture healing, anabolics

## Abstract

Aging tissues present a progressive decline in homeostasis and regenerative capacities, which has been associated with degenerative changes in tissue-specific stem cells and stem cell niches. We hypothesized that amino acids could regulate the stem cell phenotype and differentiation ability of human bone marrow-derived mesenchymal stromal cells (hBMSCs). Thus, we performed a screening of 22 standard amino acids and found that D-tryptophan (10 μM) increased the number of cells positive for the early stem cell marker SSEA-4, and the gene expression levels of *OCT-4*, *NANOG*, and *SOX-2* in hBMSCs. Comparison between D- and L-tryptophan isomers showed that the latter presents a stronger effect in inducing the mRNA levels of *Oct-4* and *Nanog*, and in increasing the osteogenic differentiation of hBMSCs. On the other hand, L-tryptophan suppressed adipogenesis. The migration and colony-forming ability of hBMSCs were also enhanced by L-tryptophan treatment. In vivo experiments delivering L-tryptophan (50 mg/kg/day) by intraperitoneal injections for three weeks confirmed that L-tryptophan significantly increased the percentage of cells positive for SSEA-4, mRNA levels of *Nanog* and *Oct-4*, and the migration and colony-forming ability of mouse BMSCs. L-kynurenine, a major metabolite of L-tryptophan, also induced similar effects of L-tryptophan in enhancing stemness and osteogenic differentiation of BMSCs in vitro and in vivo, possibly indicating the involvement of the kynurenine pathway as the downstream signaling of L-tryptophan. Finally, since BMSCs migrate to the wound healing site to promote bone healing, surgical defects of 1 mm in diameter were created in mouse femur to evaluate bone formation after two weeks of L-tryptophan or L-kynurenine injection. Both L-tryptophan and L-kynurenine accelerated bone healing compared to the PBS-injected control group. In summary, L-tryptophan enhanced the stemness and osteoblastic differentiation of BMSCs and may be used as an essential factor to maintain the stem cell properties and accelerate bone healing and/or prevent bone loss.

## 1. Introduction

Aging has been associated with a decline in the function of multiple organs due to degenerative changes in tissue-specific stem cells or stem cell niches, as well as decreased bone metabolism and osteoporosis, which are high-risk factors for bone fractures [1,2]. The current and unprecedented super-aging of the population requires urgent adaptation in the health care system worldwide. In this context, preventive approaches focusing on antiaging therapies or the treatment of age-related diseases or dysfunctions have attracted researchers to search for novel management and therapeutic approaches, including stem cell therapy. Therefore, a deeper understanding of the molecular mechanisms involved in the age-dependent deterioration of stem cell function is of critical importance for the development of new methods or materials for prevention or treatment of aging-related diseases or dysfunctions.

Amino acids are organic compounds containing amine (–NH2) and carboxylic acid (–COOH) functional groups. There are more than 500 amino acids in nature [3,4]; however, only 22 of them can be found in the human body, which are termed as 22 common amino acids, and include nine essential (need to be obtained from the diet) and 13 non-essential (synthesized in the body) amino acids [5]. Amino acids pass through the cell membrane via the amino acid transporter [6], and can be used for protein synthesis, and/or be degraded into one of seven common metabolic intermediates that can be further converted into glucose or oxidized by the citric acid cycle [7]. Additionally, amino acids play fundamental roles in processes such as neurotransmitter transport and gene transcription [8].

Amino acids are widely known as notable antiaging agents due to their abilities to promote tissue repair and cell renewal. For instance, arginine derived from nitric hoxide was proven to be essential to healthy skin [9], and glutamine regulates the body’s pH, thereby influencing the regeneration of cells and the synthesis of bone-protective antioxidants [10]. A recent report also showed that large amounts of methionine are required for the maintenance of stemness and differentiation of embryonic stem cells (ESCs) or induced pluripotent stem cells (iPSCs) [11]. Furthermore, considerable evidence suggests that the regulation of stem cell renewal and differentiation was partially modulated by the modification of metabolic and nutrient-sensing pathway [12].

On the other hand, the lack of dietary amino acids causes several symptoms or disorders such as weight loss, muscle mass decrease, decline in infection protection, and especially in changing of bone tissue homeostasis [13]. It is proven that dietary amino acids influence positively bone health by reducing bone hip fracture, promoting postfracture recovery, and increasing bone strength [14]. The impact of certain amino acids or their metabolites on typical bone diseases, age-related osteoporosis has also been reported. Arginine, for instance, was shown to support bone formation by inducing collagen and facilitating the growth of osteoblast [1,15]. In addition, a recent report has also demonstrated the importance of the kynurenine pathway during osteogenic differentiating of human mesenchymal stromal cells (hMSCs) [16].

A deeper understanding of whether amino acids can regulate both the stemness and differentiation ability towards osteogenesis and adipogenesis of bone marrow-derived stromal cells (hBMSCs) is important for future potential applications of amino acid supplementation as preventive approaches for aging-related dysfunctions of stem cell niches, particularly in the bone marrow. Therefore, we performed a screening of 22 common amino acids using human BMSCs (hBMSCs) and found that L-tryptophan could enhance stem cell markers, as well as osteogenic differentiation of BMSCs. Since L-kynurenine is the main and initial metabolite of L-tryptophan, we also investigated its effect on the cellular behavior of BMSCs in vitro and in vivo. The null hypothesis of this study was that treatment of L-tryptophan and L-kynurenine does not induce any changes in the stem cell phenotype of BMSCs.

## 2. Materials and Methods

### 2.1. BMSC Culture

Human BMSCs were purchased from Lonza (# PT-2501, Basel, Switzeland) and cultured under guidance of a previous protocol [17]. Briefly, hBMSCs were cultured with basal medium, consisting of alpha-modified minimum essential medium (α-MEM, Invitrogen, Carlsbad, CA, USA), 15% fetal bovine serum (FBS; Invitrogen), 2 mM L-glutamine (Invitrogen), 100 units/mL penicillin (Sigma-Aldrich, Saint Louis, MO, USA) and 100 mg/mL streptomycin (Sigma), at 37 °C under 5% CO_2_ in air.

mBMSCs were obtained following a previous reported protocol with minor modifications [17,18]. mBMSCs were isolated from the femur by flushing the bone marrow. The cells were collected by centrifugation and cultured in α-MEM containing 20% FBS, 1% antibiotics (penicillin 100 units/mL and streptomycin 100 mg/mL), 2 mM L-glutamine (Invitrogen), 0.1 mM L-ascorbic acid phosphate (FUJIFILM Wako Pure Chemical Corporation, Osaka, Japan) and 55 μM β-mercaptoethanol (Life Technologies, Grand Island, NY, USA). All experiments were repeated at least three times.

For pretreatment experiments with amino acid, BMSCs were cultured up to 60% confluence and further incubated with 10 μM of each amino acid for two days, and then used for subsequent analysis. BMSCs from fourth to eighth passages were used in this study. An orphan ligand library (Enzo Life Sciences, Tokyo, Japan) containing 22 common D-isoform amino acids was used for the screening (Appendix A).

### 2.2. Differentiation Assays

Osteogenesis: hBMSCs or mBMSCs were cultured up to 100% confluence, and thereafter induced to differentiate toward the osteogenic lineage by mineralization medium consisted of the respective human or mouse basal medium supplemented with 10 nM dexamethasone (Sigma-Aldrich), 2 mM β-glycerophosphate (Sigma-Aldrich), and 10 nM L-ascorbic acid phosphate, for 14 days. Cultures were then analyzed for gene expression of osteocalcin (*OCN*) and osteopontin (*OPN*) or stained for the calcified matrix with 1% Alizarin Red S (Sigma-Aldrich) staining solution [19].

Adipogenesis: hBMSCs or mBMSCs were cultured up to 100% confluence and then incubated in the respective human or mouse basal medium supplemented with 10 μg/mL insulin (Sigma), 0.5 mM 1-methyl-3-isobutylxanthine (Sigma), 60 μM indomethacin (Sigma), 0.5 μM Hydrocortisone (Sigma) for 14 days. After induction, cultures were analyzed for gene expression of Peroxisome proliferator-activated receptor gamma (PPAR-γ) and Lipoprotein lipase (LPL), or stained for oil droplets with 0.5% Oil Red O (Sigma) staining solution [20].

### 2.3. Flow-Cytometry (FCM)

Cultured hBMSCs or mBMSCs were dissociated with accutase (Inovative Cell Technologies, CA, USA) and filtered through a 70 μm cell strainer, washed, resuspended in phosphate-buffered saline (PBS) containing 1% FBS, and incubated with antihuman SSEA-4 (eBioscience, San Diego, CA, USA) for 30 min on ice. Cells were washed again and subjected to FCM analysis by Accuri C6 (BD Biosciences, Franklin Lakes, NJ, USA) [21].

### 2.4. Real-Time Reverse-Transcription Polymerase Chain Reaction (Real-Time RT-PCR)

Total cellular RNA from cells was extracted by Purelink RNA mini kit (Invitrogen) according to the manufacturer’s instructions and purified by removing genomic DNA with RNase-Free DNase set (Purelink DNase, Invitrogen). The relative levels of mRNA of target genes were normalized to that of the reference gene ribosomal protein *S29* [22]. Primer sequences are shown in Appendix A. All experiments were repeated at least three times, independently.

### 2.5. Screening Process

#### 2.5.1. First Screening

For library screening, 22 D-isoform common amino acids from an orphan ligand library were selected and used to treat hBMSCs in culture for two days at a dose of 10 µM. The cells were then harvested and analyzed by FCM for the expression of the early stem cell surface marker, SSEA-4 [23].

#### 2.5.2. Second Screening

In the second screening, three candidate molecules (D-methionine, D-proline, D-tryptophan) were selected from the first screening and used to treat hBMSCs for two days at dose of 10 µM. The total cellular RNA was extracted from the cells for analysis of the mRNA expression levels of the stem cell markers *NANOG*, *SOX-2*, and *OCT-4* [24].

### 2.6. Immunocytochemistry

A total of 1 × 10^4^ hBMSCs were seeded onto 96-well plates and cultured in the presence of amino acids for 48 h in basal medium. The cells were subsequently fixed in 4% paraformaldehyde (PFA) for 15 min, permeabilized with PBS containing 0.25% Triton X-100 for 10 min, blocked with 5% normal goat serum (Invitrogen), and then incubated with primary antibody anti-CD146 (Abcam, Cambridge, MA, USA), anti-Ki-67 (Abcam) or the respective IgG (Abcam) overnight at 4 °C. Cells were then incubated with secondary antibody Alexa Fluor 488-conjugated IgG (Invitrogen) for 1 h at room temperature, and observed under a fluorescence microscope (BZ-X700, KEYENCE, Osaka, Japan). Cell nuclei were stained with 4′6-diamidino-2-phenylindole (DAPI, Invitrogen) [25].

### 2.7. Cell Viability

A total of 1 × 10^4^ hBMSCs were seeded onto 96 well plates and treated with amino acids for 48 h. Cell viability was assessed by a colorimetric assay based on the bio-reduction of a tetrazolium compound [3-(4,5-dimethylthiazol-2-yl)-5-(3-carboxymethoxyphenyl)-2-(4-sulfophenyl)-2H-tetrazolium, inner salt; MTS] by viable cells (CellTiter 96 AQueous One Solution Cell Proliferation Assay; Promega, Madison, WI, USA), according to manufacturer’s instructions.

### 2.8. Migration Assay

Migration assay was performed according to the Boyden chamber method, using cell culture inserts with a light-opaque, polyethylene terephthalate, 8-μm, microporous membrane (BD Falcon HTS FluoroBlok inserts, BD Biosciences). BMSCs were dissociated with accutase, resuspended in basal medium and counted. Cells were then washed with PBS, centrifuged, and resuspended with serum-starved (1% FBS) medium. Cells were counted one more time, and 5000 cells contained in 250 μL of serum-starved medium were seeded in the upper chamber (cell insert), while in the lower chamber 750 μL of basal culture medium (15% FBS). After 24 h of incubation, cells were fixed with 4% PFA, stained with 20 μg/mL of Alexa Fluor 546-conjugated Phalloidin (Invitrogen) overnight, washed with PBS, and observed under a fluorescence microscope (BZ-X700, Keyence, Osaka, Japan). The total number of migrated cells observed at the bottom of the chamber was counted as an average of four different pictures taken per chamber [26].

### 2.9. Animal Experiments

Five-week-old C57BL/6 mice were used in the experiments, according to the Guidelines for Animal Research of Okayama University Dental School, under the approval of the Okayama University Ethical Committee (OKU-2013125). For analysis of the effect of L-tryptophan or L-kynurenine in vivo, mice were injected intraperitoneally with 50 mg/kg/day of L-tryptophan (Sigma-Aldrich) or L-kynurenine sulfate (Sigma-Aldrich) for consecutive three weeks, and then femurs were dissected from mice for mBMSC isolation or further micro-CT and histological analysis.

For the surgical experiment, mice were anesthetized by inhalation of isoflurane (Isoflu: Dainippon Sumitomo Pharma Co., Osaka, Japan). The left lower limb was shaved and aseptically cleaned with 70% ethanol. An incision of approximately 15 mm in the frontal skin of the mid-femur was performed to expose the muscle. The muscle was then elevated and the periosteum was separated to expose the femur surface. A drill was used to make a surgical defect of 1 mm in diameter in the anterior portion of the diaphysis, 5 mm above the knee joint [27]. The diaphysis was irrigated with saline during the surgery. Thereafter, the muscles were replaced in the original position and the incision line was sutured. 

### 2.10. Colony-Forming Assay

Colony-forming unit-fibroblast (CFU-F) assay was performed under two different protocols. In the first one, 2 × 10^6^ of mouse bone marrow cells collected from mouse femur were cultured in 6 cm^2^ culture dishes with or without L-tryptophan or L-kynurenine (10 μM) for three weeks. In the second protocol, mice were injected L-tryptophan or L-kynurenine or PBS continuously for three weeks, and then a total of 2 × 10^6^ cells collected from the mice femurs were seeded onto 6 cm^2^ culture dishes and cultured for three weeks. After the 3-week culture period, the cells were then washed with PBS and stained with 0.1% toluidine blue contained in 1% PFA overnight. On the following day, dishes were washed to remove excess dye, and only stained clusters containing more than 50 cells were counted as colonies [28].

### 2.11. Micro-Computed Tomographic Analysis

Mouse femur defects were scanned by micro-computed tomography (micro-CT) at a resolution of 6.4 μm, and the bone volume in defect site was analyzed with Skyscan 1174 version 2 software (Nrecon, CTAn, CTvol, and CTvox), as described previously [29].

### 2.12. Histological Analysis

Femurs were fixed in 4% PFA for 2–3 days, kept in 70% ethanol before being decalcified with Morse’s solution (22.5% formic acid-10% sodium citrate solution). Samples were then washed and dehydrated through a graded ethanol series and xylene before paraffin embedding. Five-micrometer sections were stained by Hematoxylin and Eosin (H&E) and observed under a microscope (Biozero BZ-X700) [21].

### 2.13. Statistical Analysis

Statistical analyses were performed with unpaired Student’s *t*-test or one-way ANOVA with Tukey correction tests. Prism GraphPad sofware version 5.0 (San Diego, CA, USA) was used for the analyses. A statistically significant difference was considered as *p* < 0.05.

## 3. Results

### 3.1. Screening Process

In the first screening, three candidates (D-methionine, D-proline, D-tryptophan) differentially increased the percentage of hBMSCs positive for SSEA-4 after 48 h of treatment (Figure 1A). Notably, however, in the second screening, gene expression analysis showed a higher increase in the mRNA levels of the stem cell markers *NANOG*, *SOX-2*, and *OCT-4* upon D-tryptophan treatment, compared to D-methionine or D-proline (Figure 1B). Therefore, D-tryptophan was selected for the subsequent steps.

In order to determine the optimal effective dosage of D-Tryptophan for hBMSCs, we analyzed four concentrations of D-tryptophan (2, 10, 50, and 100 μM). As shown in Figure 1C, 10 μM of D-tryptophan yielded the highest increase in mRNA levels of *NANOG*, *OCT-4*, and *SOX2* in hBMSCs. This result was also confirmed by FCM analysis, which showed the highest increase in the number of cells positive for SSEA-4 upon treatment with 10 μM of D-tryptophan (Figure 1D). Therefore, we selected 10 μM as the optimal concentration for our cells.

Tryptophan is an essential amino acid that needs to be taken from the diet. It exists in two mirror forms, i.e., L- and D- isomers. Therefore, we compared the effect of the two isomers on the stem cell phenotype of hBMSCs. As shown in Table 1, L-tryptophan showed similar effects compared to D-isomer in increasing the number of SSEA-4 positive cells and the gene expression levels of *NANOG*, *SOX2*, and *OCT-4* in hBMSCs. Additionally, immunofluorescence staining demonstrated an increase in the expression of CD146, which is one major marker of BMSCs, after L-tryptophan treatment (Appendix A). Based on these results, and taking into consideration that L-tryptophan is the most common isomer in natural food and the isomer used for protein synthesis in the body [30], L-tryptophan was then selected for the subsequent experiments, at a concentration of 10 μM.

### 3.2. L-Tryptophan Enhances Migration, Colony Formation and Osteogenic Differentiation of hBMSCs In Vitro

We then investigated the effect of L-tryptophan on other cellular activities of BMSCs, including colony formation, cell migration, and differentiation towards osteogenic and adipogenic lineages. Interestingly, treatment with L-tryptophan significantly enhanced the formation of mBMSC colonies (Figure 2A). Next, we examined the effect of L-tryptophan on cell migration and proliferation. As shown in Figure 2B, L-tryptophan-treated hBMSCs exhibited a higher ability for migration than the control group; however, L-tryptophan treatment induced no significant changes in the number of cells positive for the cell proliferation marker Ki-67 (Appendix A), or in the cell proliferation (viability) analyzed by MTS assay (Appendix A).

Regarding the differentiation ability of hBMSCs, L-tryptophan enhanced osteogenesis as demonstrated by stronger staining of the mineralized matrix with Alizarin Red S (Figure 2C); as well as by an increase in mRNA levels of osteogenic markers *OPN* and *OCN* (Figure 2D). Interestingly, L-tryptophan suppressed adipogenesis as can be seen in Oil Red O staining of cell cultures, and decreased mRNA levels of adipogenic markers *PPARγ* and *LPL* (Figure 2E,F).

### 3.3. L-Tryptophan Enhances Migration, Colony Formation and Osteogenic Differentiation of mBMSCs In Vivo

Next, in order to analyze the effect of L-tryptophan in vivo, we injected 10 mg/kg/day or 50 kg/mg/day of L-tryptophan in mice intraperitoneally for consecutive three weeks. Mice femurs were then dissected for isolation of mBMSCs in culture dishes for subsequent assays. After 21 days of in vitro culture, mBMSCs were submitted to FCM analysis, which showed that the concentration of 50 mg/kg/day induced a significant increase in the number of SSEA-4 positive cells, whereas 10 mg/kg/day induced no significant changes (Figure 3A). Consistently, L-tryptophan (50 kg/mg/day) increased the mRNA levels of *Nanog* and *Oct-4* (Figure 3B) and the colony-forming ability of mBMSCs, compared to the PBS-injected control group (Figure 3C). We also examined the differentiation ability of mBMSCs toward osteogenesis and adipogenesis. Consistent with the in vitro results with hBMSCs, L-tryptophan enhanced the deposition of the mineralized matrix as shown by Alizarin Red S staining (Figure 3D), as well as the mRNA levels of *Opn* and *Ocn* (Figure 3E). On the other hand, there was no significant difference in the adipogenic differentiation of mBMSCs after L-tryptophan injection, as analyzed by Oil Red O staining and mRNA levels of *Pparγ* and *Lpl* (Figure 3F,G). Taken together, these results indicate that L-tryptophan can promote an increase in stemness and osteogenic ability of BMSCs in vitro and in vivo.

### 3.4. L-Kynurenine, A Main Metabolite of L-Tryptophan, Enhances the Stem Cell Phenotype of hBMSCs

Previous investigations reported that 99% of L-tryptophan is catalyzed by the enzymes tryptophan 2,3 dioxygenase (TDO) and Indoleamine 2, 3-dioxygenase (IDO1) and subsequently metabolized through the L-kynurenine pathway [30,31]. Therefore, since L-kynurenine is the main metabolite of L-tryptophan, we evaluated the effects of L-kynurenine on the stem cell phenotype and differentiation ability of hBMSCs. As expected, in vitro analyses showed that L-kynurenine induced similar effects of those promoted by L-tryptophan, including an increase in the number of cells positive for SSEA-4 (Figure 4A), in gene expression levels of *NANOG*, *SOX2*, and *OCT-4* (Figure 4B), as well as in the ability of mBMSCs to form colonies (Figure 4C) and to migrate (Figure 4D). Figure 4E shows that L-kynurenine also enhanced osteogenesis of hBMSCs as demonstrated by stronger staining of the calcified matrix with Alizarin Red S, and the increase in *OPN* and *OCN* mRNA levels. On the other hand, no significant changes in adipogenic differentiation of hBMSCs were observed upon L-kynurenine treatment, as analyzed by Oil Red O staining of oil droplets, and gene expression levels of *PPARγ* and *LPL* (Figure 4H).

Taken together, these results suggest that L-kynurenine is the major pathway associated with the effect of L-tryptophan on promoting the increase in stemness and osteogenic ability of BMSCs.

### 3.5. Both L-Tryptophan and L-Kynurenine Accelerate Bone Regeneration in Mouse Femur Surgical Defect

We have previously demonstrated that during the inflammatory phase of wound healing, the inflammatory cytokine, tumor necrosis factor-α (TNF-α), enhanced the stem cell phenotype of MSCs [20]. Additionally, we also showed that MSCs intensively migrate to the wound site during the inflammatory phase of healing (initial two to three days post-trauma) to accelerate bone healing [20,32]. Based on these previous data, as well as on the fact that L-tryptophan and L-kynurenine enhanced the stemness and migration ability of BMSCs, we hypothesized that in vivo treatment with the two amino acids could also enhance bone healing. Therefore, we used a mouse model of femur surgical defect and administrated L-tryptophan or L-kynurenine (50 mg/kg/day) or PBS by intraperitoneal injections one week before the surgical defect was made, and subsequently for two weeks after the surgery (Figure 5A). As demonstrated by micro-CT analysis and quantitative measurements, we observed an enhanced bone regeneration both in L-tryptophan (Figure 5B) and L-kynurenine-treated groups (Figure 5D), compared to their respective PBS-injected control group. Moreover, histological sections further confirmed that L-tryptophan and L-kynurenine accelerated new bone formation (Figure 5C,E).

## 4. Discussion

The involvement of amino acids in aging and bone homeostasis, as well as in the maintenance and differentiation of iPS cells as outlined in recent reports, strongly supports the notion that amino acids are essential for the regulation of maintenance and differentiation of adult stem cells. In our screening of over 22 amino acids, tryptophan was shown to differentially enhance the expression of stem cell markers in hBMSCs in vitro, and also maintain the stemness of BMSCs in vivo. Additionally, we demonstrated that L-tryptophan can enhance osteogenesis of BMSCs and bone regeneration in vivo. These results are in accordance with previous reports showing that the tryptophan-free diet decreased body weight and delayed femoral bone growth and bone mineral density in rats [33]. Therefore, tryptophan has crucial roles in the maintenance of stemness of BMSCs and bone homeostasis.

The in vitro and in vivo results showing enhanced stemness of BMSCs and enhanced bone formation upon L-tryptophan stimulation may seem contradictory. However, since BMSCs consist of a heterogenic population, the effect of L-tryptophan could be both on stem/progenitor cells and osteoblastic cells. Moreover, the enhanced stem cell phenotype of BMSCs by the amino acid treatment could, in a latter step, promote a stronger differentiation of the cells. However, it is not clear from the present results whether the effect of L-tryptophan on inducing no significant changes or an inhibition of the adipogenic differentiation of BMSCs was naturally due to a direct effect of the amino acid in suppressing the expression of master genes of adipogenic differentiation (e.g., PPARγ) or an indirect effect due to activation of the master genes of osteoblastic differentiation (e.g., RUNX2), which, consequently, would activate the osteoblast-adipocyte switch towards the osteoblastic lineage. Future studies investigating the possible regulation of this molecular switch by amino acids may provide a deeper understanding of the mechanisms of L-tryptophan associated inhibition of adipogenic differentiation.

L-tryptophan passes through the cell membrane via the L-amino acid transporter system, which is the major route for providing the living cells with essential amino acids [6]. Tryptophan can be then metabolized into two main pathways (serotonin or kynurenine pathways). It has been reported that serotonin has either an inhibitory or stimulatory effect on bone growth, depending on whether it originated from the gut or the central nervous system (acting as a neurotransmitter), respectively [34,35,36]. Importantly, only 1% of tryptophan is metabolized into the serotonin pathway, while the majority of 99% of tryptophan is degraded through the kynurenine pathway [31,37] by the TDO and IDO1 enzymes. In addition, in in vitro assays of osteogenic differentiation of hBMSCs, no changes in serotonin levels in the supernatant of hBMSCs culture medium could be detected, while the kynurenine pathway was shown to be strongly activated [16].

In this context, we considered L-kynurenine as the major metabolite of tryptophan and then proceeded with the next experiments investigating the effects of L-kynurenine on the stemness of hBMSCs. As expected, we found that L-kynurenine, similar to L-tryptophan, enhanced the stem cell phenotype of BMSCs in vitro and in vivo. These findings are also in agreement with those of a recent study using cancer cells, that reported that L-kynurenine acts as an endogenous ligand of the ligand-activated transcription factor aryl hydrocarbon receptor (AhR) and can subsequently activate the transcription level of the master pluripotency factor *Oct-4* [8].

Additionally, in accordance with previous reports, we also demonstrated that L-kynurenine enhances osteogenesis and bone formation in vivo [16]. A previous clinical epidemiological study also found high serum levels of kynurenine to be associated with lowerer BMD and risk of hip fracture in elderly individuals [38,39]. Importantly, the deletion of IDO-1, the major enzyme degrading tryptophan to kynurenine, markedly reduced the osteoblast number and bone volume in the knockout mice [16], indicating, therefore, the potential role of kynurenine as the major downstream pathway of tryptophan for the maintenance of bone homeostasis.

Regarding the process of stem cell aging, recent evidence indicated that the maintenance of stem cell function and differentiation is modulated by mTORC1 pathway or by the modification of the nutrient-sensing pathways, such as AMPK [12]. In an attempt to find other cellular pathways that could be associated with the increase in stemness of hBMSCs, we performed a pathway array and an RNA array for stem cell signaling genes; however, we could identify no molecules involved in the upregulation of stem cell markers in hBMSCs (data not shown). Future studies using different approaches or analyzing changes at a single cell level may provide a deeper understanding of the intracellular mechanisms of amino acid-regulation of stemness.

## 5. Conclusions

In summary, our data provide evidence of the novel effect of L-tryptophan and L-kynurenine in increasing the stem cell phenotype of BMCSs, which could further implicate in the maintenance of bone homeostasis. Further understanding of the molecular signaling involved in these effects may also elucidate the processes underlying stem cell aging. Finally, L-tryptophan and/or L-kynurenine could be potential targets for the development of novel materials and therapeutics for the aging-related decline in stem cell properties of tissue-specific stem/progenitor cells, and bone loss, including osteoporosis.

## Figures and Tables

**Figure 1 materials-14-00208-f001:**
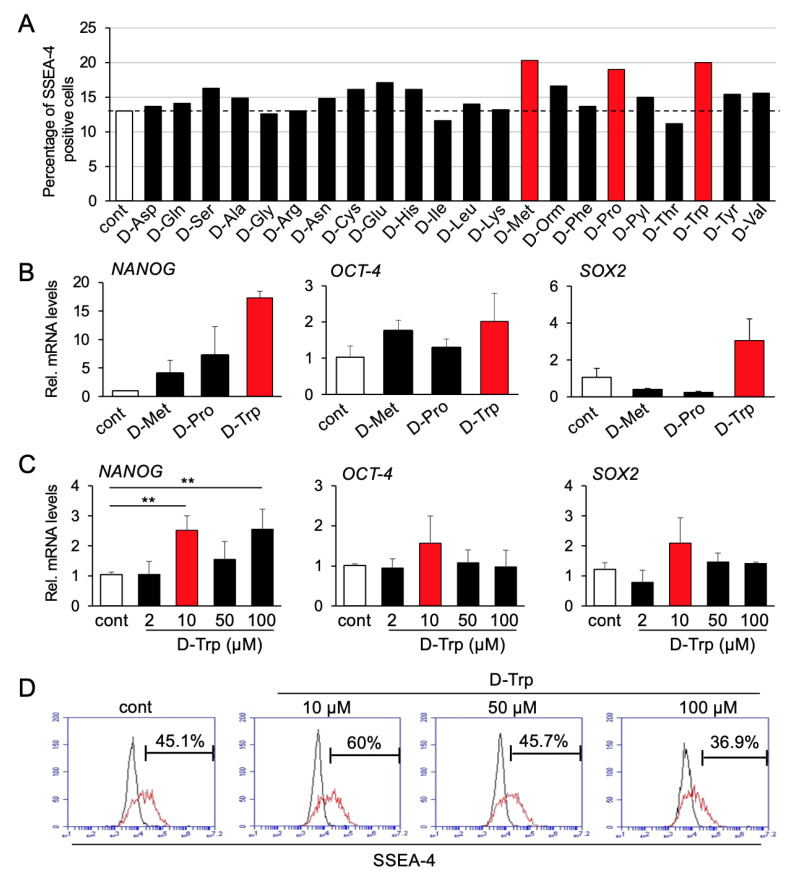
Screening process. (**A**) First screening by flow cytometry. Three candidate amino acids were selected (red columns): D-methionine (D-Met), D-proline (D-Pro), and D-tryptophan (D-Trp). The experiment was performed once. (**B**) Second screening by real-time RT-PCR. D-tryptophan (D-Trp) strongly enhanced the mRNA levels of stem cell markers, *NANOG*, *OCT-4*, and *SOX-2*. Results are representative of two independent experiments. (**C**,**D**) hBMSCs were treated with four different concentrations (2, 10, 50, and 100 μM) of D-Trp for 48 h, and 10 μM concentration induced the highest increase in gene expression levels of *NANOG*, *OCT-4* and *SOX2* (**C**) and the number of SSEA-4 positive cells (**D**). Results are representative of at least two independent experiments. (** *p* < 0.01, one-way ANOVA/Tukey).

**Figure 2 materials-14-00208-f002:**
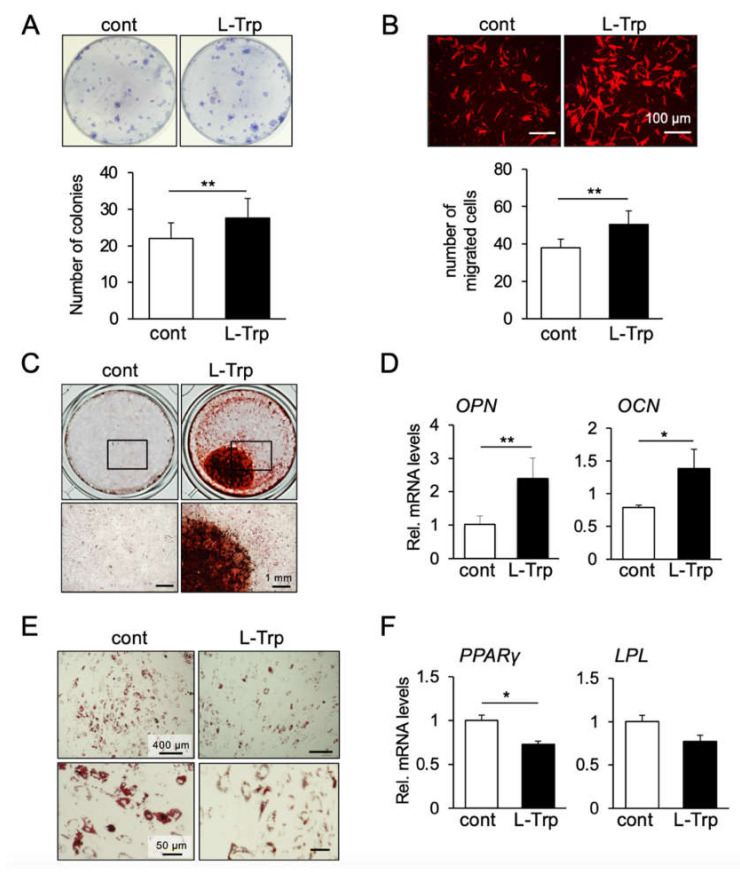
Effect of L-tryptophan on the stemness-related function and differentiation capability of hBMSCs in vitro. (**A**) CFU-F assay was performed using primary mBMSCs. The colony-forming potential was increased by the treatment of L-Trp. Data represent the means ± SD (n = 3). ** *p* < 0.01, unpaired t-test. (**B**) L-Trp induced higher migration ability compared to the control group. The images are representatives of three independent experiments. Graph shows the average number of migrated cells counted in four different pictures taken at each chamber. ** *p* < 0.01, unpaired t-test. Pre-treatment of L-Trp induced osteogenic differentiation of hBMSCs (**C**,**D**). L-Trp decreased the adipogenic ability of hBMSCs (**E**,**F**). * *p* < 0.05, ** *p* < 0.01, unpaired t-test. Images and graphs are representatives of at least three independent experiments.

**Figure 3 materials-14-00208-f003:**
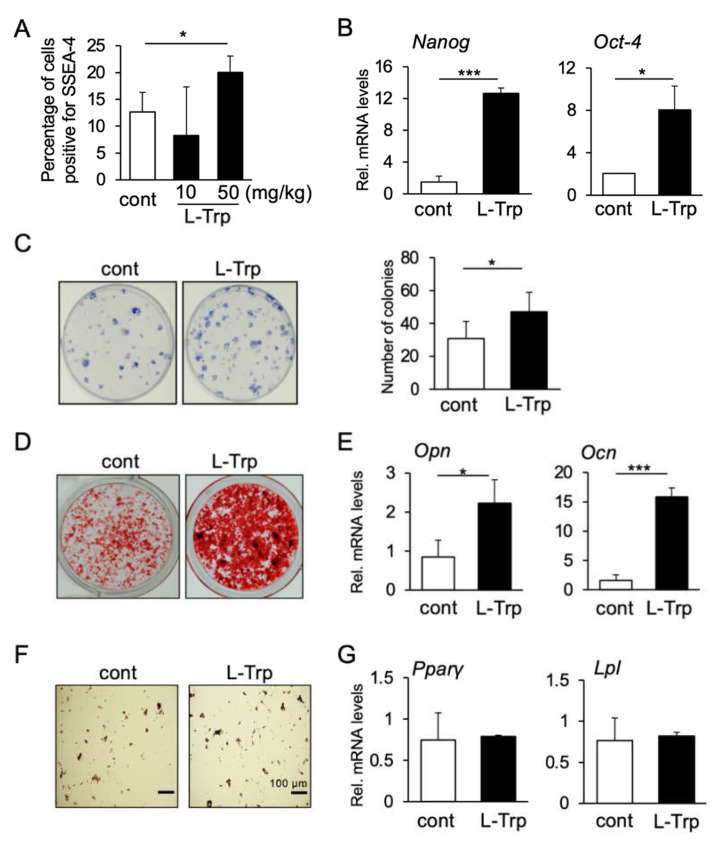
Effect of L-tryptophan on the stem cell phenotype of mBMSCs and trabecular bone in vivo. (**A**) FCM analysis of the percentage of mBMSCs positive for SSEA-4. mBMSCs from L-Trp (50 mg/kg)-treated mice showed a significantly higher number of SSEA-4 positive cells than the control group. * *p* < 0.05, one-way ANOVA/Turkey. (**B**) Gene expression levels of *Nanog* and *Oct-4* in mBMSCs from 50 mg/kg L-Trp-treated mice was significantly higher than mBMSCs from the PBS-injected control group. Data represent the mean ± SD (n = 3). * *p* < 0.05, *** *p* < 0.001, unpaired *t*-test. (**C**) Treatment with L-Trp increased the colony-forming ability of mBMSCs. Data represent the mean ± SD (n = 3). * *p* < 0.05, unpaired *t*-test. (**D**,**E**) Differentiation ability. L-Trp induced stronger deposition of the calcified matrix as detected by Alizarin Red S staining (**D**) as well as an increase in the mRNA levels of *Opn* and *Ocn* on mBMSCs (**E**). No substantial change in Oil Red O staining (**F**) or gene expression levels of *Ppar-γ* or *Lpl* (**G**) could be observed upon L-Trp treatment. * *p* < 0.05, *** *p* < 0.001, unpaired test. Images and graphs are representative of at least three independent experiments.

**Figure 4 materials-14-00208-f004:**
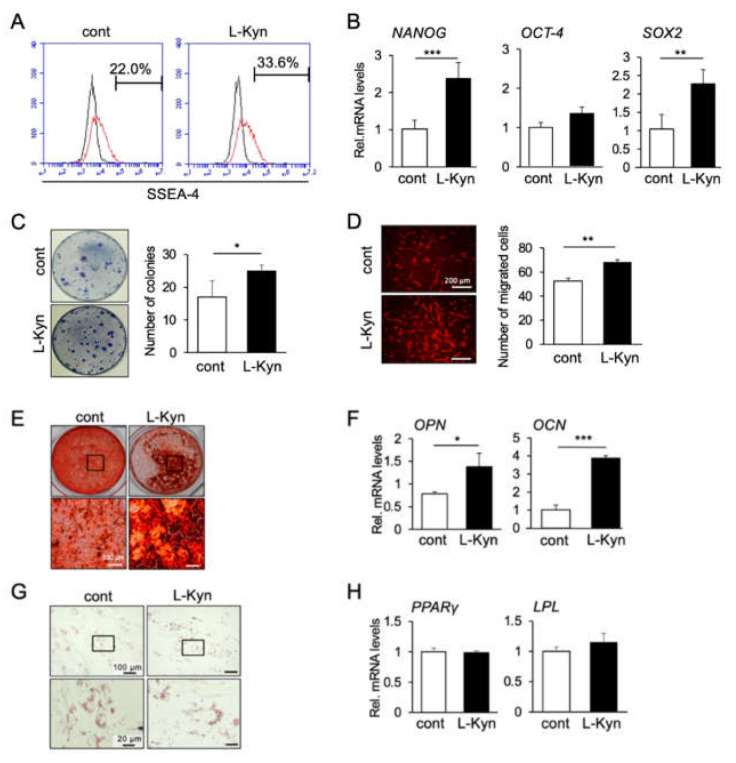
Effect of L-kynurenine on the stemness, function and differentiation capability of human bone marrow-derived mesenchymal stromal cells in vitro. (**A**) Flow cytometry showed the cells cultured with L-Kyn had a higher percentage positive with SSEA-4^+^ cells compared to control samples. (**B**) L-Kyn increased the gene expression levels of *NANOG*, *OCT-4*, *SOX2*. ** *p* < 0.01, *** *p* < 0.001, unpaired *t*-test. Data are representative of at least three independent experiments. (**C**) A higher number of colonies was observed in L-Kyn-treated group. Data represent the mean ± SD (n = 3). * *p* < 0.05, unpaired *t*-test. (**D**) hBMSCs treated with L-Kyn presented higher migration ability. Images are representative of three independent experiments. Scale bar = 100 μM. Graph shows the average number of migrated cells counted in four different pictures taken in each chamber. ** *p* < 0.01, unpaired *t*-test. (**E**,**F**) L-Kyn-treated hBMSCs presented stronger deposition of the calcified matrix as detected by Alizarin Red S staining (**E**) as well as an increase in the mRNA levels of *Opn* and *Ocn* (**F**). No change in adipogenesis of hBMSC was observed after treatment with L-Kyn (**G**,**H**). * *p* < 0.05, *** *p* < 0.001, unpaired *t*-test. Images and graphs are representative of at least three independent experiments.

**Figure 5 materials-14-00208-f005:**
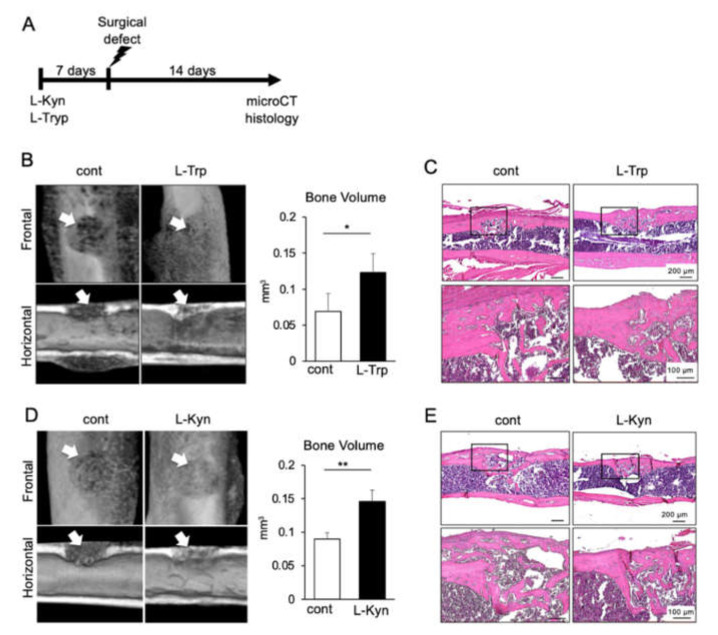
Effect of tryptophan or kynurenine on bone regeneration in mice femur surgical defect. (**A**) Schema showing the study design. Mice were then administered the two amino acids for 2 more weeks. (**B**,**D**) Three-dimensional micro-CT images of mouse femur defects after intraperitoneal injection with PBS or L-Trp (50 mg/kg) or L-Kyn (50 mg/kg). The bone healing ability after three weeks of L-Trp or L-Kyn-treated group was accelerated compared to the respective PBS-injected control group. The analysis was performed with Skyscan software. The images are representative of four different samples. The white arrow indicates the defect. (**C**,**E**) Histological analysis of mouse femurs in the defect area. Note the amount of regenerated bone volume in L-Trp (**C**) and L-Kyn-treated (**E**) groups. Data represent the mean ± SD (n = 4). * *p* < 0.05, ** *p* < 0.01, unpaired *t*-test.

**Table 1 materials-14-00208-t001:** Comparison effect of L- and D-tryptophan. Both amino acids induced an increase in the percentage of SSEA-4 positive cells (FCM analysis) and mRNA levels of stem cell markers as *NANOG*, *SOX2*, *OCT-4*, compared to the control group. The value is the ratio between treated and control groups. The results are representative of at least three independent experiments. * *p* < 0.05, ** *p* < 0.01, *** *p* < 0.001, ns: non-significant, unpaired *t*-test.

	Amino Acid	D-Trp (10 μM)	L-Trp (10 μM)
Stem Cell Marker			
SSEA-4+ cells		1.39 ± 0.03 (***)	1.21 ± 0.08 (*)
*NANOG* mRNA levels		2.25 ± 0.59 (***)	3.57 ±0.96 (***)
*OCT-4* mRNA levels		1.23 ± 0.43 (NS)	1.78 ± 0.86 (NS)
*SOX2* mRNA levels		1.45 ± 0.34 (*)	2.74 ± 1.07 (**)

## Data Availability

The data presented in this study are available on request from the corresponding authors.

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
