# Peer review of "Tryptophan and Kynurenine Enhances the Stemness and Osteogenic Differentiation of Bone Marrow-Derived Mesenchymal Stromal Cells In Vitro and In Vivo"

_materials, 2021, doi:10.3390/ma14010208_

Round 1

Reviewer 1 Report

Dear Authors, thank you for your work which I found very interesting. Howver I can see that it deals with the effect of two amminoacids on the osteogenic differentiation of hBMSCs. Accordingly no MATERIAL has been tested on your work. Therefore I think that, despite valuable, your manuscript doesn’t fit the journal’s topic and should be submitted elsewhere. By the way, I give you some advice to improve ypur work.

ABSTRACT

I see that you wrote “NANOG …OCT-4” and “Nanog….Oct-4”. PLease write in the same way

INTRODUCTION

The first ten lines of the introduction should have one or two references.

“.has also been reported Arginine…”: please add a point between reported and arginine

At the end of the introduction please add the null hypothesis of the study

MATERIALS AND METHODS

2.1.

Lonza: please add the city and country of the Manufacturer

Please add some references for the method you used for BMSC culture. Experiments should be reproducible!

2.7

Please add references

2.11

Please add references

2.12

Please add the software used for statistical analysis (with city and country)

RESULTS

3.1.

I think this section should be divided into two parts: one bleonging to Materials and Methods and the other to Materials.

In all the subsections of results please remove the description of what you have done. That is part of materials and methods, not of Results. It is a repetition.

Author Response

ABSTRACT

Comment #1: I see that you wrote “NANOG …OCT-4” and “Nanog….Oct-4”. PLease write in the same way

 Response #1: The authors thank the reviewer’s comment. According to the guideline of the international committee on genetic symbol and nomenclature, the human gene is written all in capital, and mouse gene is written with only the first letter in uppercase. Here we used NANOG and OCT-4 for human BMSCs, and used Nanog and Oct-4 for mouse BMSCs.

Reference: “Report of the International Committee on Genetic Symbols and Nomenclature (1957). Union of International Sci Biol Ser B, Colloquia No. 30.”

INTRODUCTION

Comment #2: The first ten lines of the introduction should have one or two references.

Response #2: The authors greatly thank the reviewer ‘s suggestion and have added 2 more references to the first paragraph in the revised manuscript.

Comment #3: “.has also been reported Arginine…”: please add a point between reported and arginine

Response #3: The authors thank the reviewer’s comment, and have added a point between “reported” and “Arginine”.

Comment #4: At the end of the introduction please add the null hypothesis of the study

 Response #4: The authors thank the reviewer’s suggestion, and have included the null hypothesis of the study at the end of the introduction.

MATERIALS AND METHODS

2.1.

Comment #5: Lonza: please add the city and country of the Manufacturer

Response #5: The authors thank the reviewer’s comment, and have added the information of the city and country of the manufacturer.

Comment #6: Please add some references for the method you used for BMSC culture. Experiments should be reproducible!

Response #6: The authors greatly thank the reviewer ‘s suggestion and have added a reference to the paragraph in the revised manuscript.

2.7

Comment #7: Please add references

Response #7: The authors greatly thank the reviewer ‘s suggestion and have added a reference to the paragraph in the revised manuscript.

2.11

Comment #8: Please add references

Response #8: The authors greatly thank the reviewer ‘s suggestion and have added a reference to the paragraph in the revised manuscript.  

2.12

Comment #9: Please add the software used for statistical analysis (with city and country)

Response #9: The authors greatly thank the reviewer ‘s comment and have added information of city and country of the statistical analysis software to the paragraph in the revised manuscript.

RESULTS

3.1.

Comment #10: I think this section should be divided into two parts: one belonging to Materials and Methods and the other to Materials.

Response #10: The authors greatly thank the reviewer ‘s suggestion and have divided this section into 2 parts: One belongs to the section of Material and Method ( section 2.4 page 3), and the other part we maintained in the Result (page 5, first paragraph) section of the revised manuscript.  

Comment #11: In all the subsections of results please remove the description of what you have done. That is part of materials and methods, not of Results. It is a repetition.

Response #11: The authors greatly thank the reviewer ‘s suggestion and have removed all the repeated descriptions in the subsection of result in the revised manuscript.  

Reviewer 2 Report

The authors have presented their work in using amino acids, L-Tryptophan and L-Kynurenine, to promote stemness and osteogenicity, with both in vitro experiments and in vivo support of their hypotheses. The paper is well organized and the reviewer appreciates the clarity of the presentation of the introduction and discussion. This manuscript merits publication in this journal after addressing the following concerns: - FIGURE 1 is missing! Please incorporate Figure 1 for proper review. - Text formatting is inconsistent. Figure legends should be the same size as the figure legend title. Currently it looks like figure legends are paragraphs of the results sections. - Also, in sections 3.2 and 3.3, some paragraphs are center-aligned, which should be left-aligned or justified.

Author Response

Comment #1: FIGURE 1 is missing! Please incorporate Figure 1 for proper review.

Response #1: The authors greatly thank the reviewer ‘s observation and apologize for the missing of Figure 1, and have added the Figure 1 to this section in the revised manuscript. Of note, the initial file uploaded by the authors contained Fig. 1, but it was probably unintentionally removed during manuscript editing by MDPI.

Comment #2: Text formatting is inconsistent. Figure legends should be the same size as the figure legend title. Currently it looks like figure legends are paragraphs of the results sections.

Response #2: The authors thank the detailed observation made by the reviewer, and have edited the font size in the figures legend.

Comment #3: Also, in sections 3.2 and 3.3, some paragraphs are center-aligned, which should be left-aligned or justified.

Response #3: The authors thank the detailed observation made by the reviewer, and have aligned the paragraph in the revised manuscript.

Reviewer 3 Report

This paper aims to describe the role played by the L-Tryptophan (and its metabolite L-Kynurenine) in the maintenance of stemness of BMSCs and bone homeostasis. In the paper, authors demonstrate that Tryptophan is able to differentially enhance the expression of stem cell markers in hBMSCs in vitro, and also maintain the stemness of BMSCs in vivo. Moreover, they demonstrate that L-Tryptophan can enhance osteogenesis of BMSCs and bone regeneration in a rat model of bone healing in vivo.

The novelty of the work resides in that authors have seen the effect of the aminoacid, and its principal metabolite. It also demonstrates that amino acids are essential for the regulation of maintenance and differentiation of adult stem cells.

The paper is well written and the methods are clearly explained to be reproduced. 

Major comments:

  • Figure 1 is not present in the manuscript. This reviewer has not had the possibility to contrast the data.
  • In order to really demonstrate that the effects played by L-Tryptophan in the BMSCs are mediated through its conversion into L-Kynurenine, authors may employ specific chemical or biological inhibitors (siRNA) of the Tryptophan 2,3 dioxygenase (TDO) and Indoleamine 2, 3-dioxygenase (IDO1), and see if they prevent the effects described by the aminoacid. To date, what they present in the paper could be seen as a “correlation of effects”.
  • Please, provide precise references for the gene markers employed.
  • The mRNA levels of the markers are important. However, authors could validate the data presenting western bot studies of the protein markes.

Minor points:

  • Authors should consider to move the results involving the adipogenesis and the adipogenic markers (Fig 2, 3 and 4) to supplementary data.
  • Authors should consider to remove the data from the 10 kg/mg/day of L-Tryptophan in Figure 3. They do not provide substantial data.
  • Authors should consider to rephrase the Title “3.3. L-Tryptophan enhances stem cell phenotype of mBMSCs in vivo”. They do not see differences in the adipogenic differentiation, so they can employ the same title of 3.2 , where they employ “osteogenic differentiation”.
  • Table 1. Please, explain the values of the table. Are the values of the table referred to the controls? .
  • Please, provide a reference for the statement “that there is 22 standard aminoacids”. Traditionally it has been described 20 standard aminoacids. G-D-Glutamylglycine and D-Pyroglutamic acid are not considered standard aminoacids to the knowledge of this referee.
  • In some cases, authors mention L-tryptophan in the text. Please employ L-Tryptophan.
  • Page 2 line “osteoporosis has also been reported Arginine, for instance, was shown “. There is a point missing between “reported and Arginine”.
  • Page 3 line “Cultured hBMSCs or mBMSCs were dissociated with accutase”. Please, include the Company name and country of the product accutase.
  • Page 3 line “Total cellular RNA from cells was extracted by Purelink RNA mini kit (Invitrogen) according to the manufacturer's instructions and purified by removing genomic DNA with RNase-Free DNase set (Purelink DNase, Invitrogen). The relative levels of mRNA of target genes were normalized to that of the reference gene ribosomal protein S29 [17]. Primer sequences are shown in Supplementary Table S2. All experiments were repeated at least three times, independently” . Please, check the size of the letters in those sentences.
  • Page 4 line “Immunocytochemistry studies”. Please, include the number of cells seeded in the 96 well.
  • Page 3 line: 2.11. Histological analysis. Please, include the concentrations of the formic acid-sodium citrate solution.
  • Authors should consider to reorganize the letters in Figures 2, 3 and 4 so that figures follow the normal order, starting from the left to the right, from the top to the bottom.
  • Page 8. The text of paragraphs from the 3.3 are not justified in both sides.
  • Figure 4. Please, check the figure footnote style.

Author Response

Major comments:

Comment #1: Figure 1 is not present in the manuscript. This reviewer has not had the possibility to contrast the data.

Response #1: The authors greatly thank the reviewer ‘s observation and apologize for the missing of Figure 1, and have added the Figure 1 to this section in the revised manuscript. Of note, the initial file uploaded by the authors contained Fig. 1, but it was probably unintentionally removed during manuscript editing by MDPI.

Comment #2: In order to really demonstrate that the effects played by L-Tryptophan in the BMSCs are mediated through its conversion into L-Kynurenine, authors may employ specific chemical or biological inhibitors (siRNA) of the Tryptophan 2,3 dioxygenase (TDO) and Indoleamine 2, 3-dioxygenase (IDO1), and see if they prevent the effects described by the aminoacid. To date, what they present in the paper could be seen as a “correlation of effects”.

Response #2. The authors thank the detailed comment addressed by the reviewer, and have changed the manuscript and abstract to mention the possible involvement of Kynurenine pathway in the effect of Tryptophan inducing stemness phenotype and osteogenic differentiation of BMSCs. The suggestion would be a great suggestion for our future studies to clarify the roles of Kynurenine pathway in mediating stem cell phenotype and osteogenic differentiation ability of BMSCs.

Comment #3: Please, provide precise references for the gene markers employed.

Response #3: The authors greatly thank the reviewer’s suggestion and have added references to the section 2.5 in Materials and Method in the revised manuscript.

Comment #4: The mRNA levels of the markers are important. However, authors could validate the data presenting western bot studies of the protein markes.

Response #4: The authors greatly thank the reviewer’s suggestion. In order to investigate the effect of Tryptophan and Kynurenine on the stem cell phenotype of BMSCs, we performed real time PCR (gene level) analysis to check the gene expression levels, and confirmed the expression of other stem cell markers by flow cytometry (protein level). Moreover, we carried out Alizarin red S and Oil red O staining which are commonly used to identify the calcium contained osteoblast and fat cells, respectively. Therefore, we believe that our current data are enough to support our hypothesis. We greatly appreciate the suggestion from reviewer and would appreciate if the reviewer could accept our response.

Minor points:

Comment #5: Authors should consider to move the results involving the adipogenesis and the adipogenic markers (Fig 2, 3 and 4) to supplementary data.

Response #5: The authors greatly thank the reviewer for the suggestion. We investigated the effect of L-Tryptophan and L-Kynurenine on the differentiation abilities of BMSCs. Interestingly, our data showed that the two compounds only enhanced the osteogenic differentiation, but unexpectedly, not the adipogenic differentiation of BMSCs. This finding is particularly interesting because the osteogenic and adipogenic differentiation pathways are known to be opposite pathways. Thus, the fact that the two compounds could not induce adipogenic differentiation of BMSCs may potentially be associated with new mechanisms of stem cell regulation. The authors would like to pursue this issue in the future, and therefore, we would prefer to keep the adipogenic differentiation data in the current figures.

Comment #6: Authors should consider to remove the data from the 10 kg/mg/day of L-Tryptophan in Figure 3. They do not provide substantial data.

Response #6: The authors greatly thank the reviewer for the comment. First we tested 2 different concentrations referred from the previous reports, and found a clear and remarkable effect of L-Tryptophan at a concentration of 50 kg/mg/day. The data using 10 kg/mg/day of L-Tryptophan is necessary to show that L-Tryptophan is not effect at this dosage. Therefore, we would like to show that data to explain that we have the optimized concentration.

Comment #7: Authors should consider to rephrase the Title “3.3. L-Tryptophan enhances stem cell phenotype of mBMSCs in vivo”. They do not see differences in the adipogenic differentiation, so they can employ the same title of 3.2 , where they employ “osteogenic differentiation”.

Response #7: The authors thank the detail comment addressed by reviewer and have rephrased the title of subsection 3.2 in the revised manuscript.

Comment #8: Table 1. Please, explain the values of the table. Are the values of the table referred to the controls? .

Response #8: The authors thank the comment addressed by reviewer and have added the explanation to the subsection of the table. “The value is the ratio between the treated and control groups”

Comment #9: Please, provide a reference for the statement “that there is 22 standard aminoacids”. Traditionally it has been described 20 standard aminoacids. G-D-Glutamylglycine and D-Pyroglutamic acid are not considered standard aminoacids to the knowledge of this referee.

Response #9: The authors thank the detail comment addressed by reviewer and have replaced the word “standard” by “common” and added a reference in the Introduction section (paragraph 2, line 5, page 2)

Comment #10: In some cases, authors mention L-tryptophan in the text. Please employ L-Tryptophan.

Response #10: The authors thank the comment addressed by reviewer and have edited all the errors in the revised manuscript.

Comment #11: Page 2 line “osteoporosis has also been reported Arginine, for instance, was shown “. There is a point missing between “reported and Arginine”.

Response #11: The authors thank the reviewer’s comment, and have added a point between “reported” and “Arginine”.

Comment #12: Page 3 line “Cultured hBMSCs or mBMSCs were dissociated with accutase”. Please, include the Company name and country of the product accutase.

Response #12: The authors thank the reviewer’s comment, and have included the company and country of the product.

Comment #13: Page 3 line “Total cellular RNA from cells was extracted by Purelink RNA mini kit (Invitrogen) according to the manufacturer's instructions and purified by removing genomic DNA with RNase-Free DNase set (Purelink DNase, Invitrogen). The relative levels of mRNA of target genes were normalized to that of the reference gene ribosomal protein S29 [17]. Primer sequences are shown in Supplementary Table S2. All experiments were repeated at least three times, independently”. Please, check the size of the letters in those sentences.

Response #13:  The authors thank the detailed observation made by the reviewer, and have edited the font size in this paragraph.

Comment #14: Page 4 line “Immunocytochemistry studies”. Please, include the number of cells seeded in the 96 well.

Response #14:  The authors thank the detailed observation made by the reviewer, and have included the number of cell seeded in the 96 well plates (10000 cells/ well).

Comment #15: Page 3 line: 2.11. Histological analysis. Please, include the concentrations of the formic acid-sodium citrate solution.

Response #15: The authors thank the detailed observation made by the reviewer, and have included the concentration of the reagent (22.5% formic acid-10% sodium citrate solution).

Comment #16: Authors should consider to reorganize the letters in Figures 2, 3 and 4 so that figures follow the normal order, starting from the left to the right, from the top to the bottom.

Response #16:  The authors thank the detailed observation made by the reviewer, and have reorganized the Figure.

Comment #17: Page 8. The text of paragraphs from the 3.3 are not justified in both sides.

Response #17:  The authors thank the detailed observation made by the reviewer, and have aligned the paragraphs.

Comment #18: Figure 4. Please, check the figure footnote style.

Response #18:  The authors thank the detailed observation made by the reviewer, and have edited the footnote style.

Round 2

Reviewer 1 Report

Dear Authors,

thank you for your revised version which I found much more adequate. However, I have a concern as regards the topic of your research which might fit much more another journal. In fact, no "MATERIAL" has been tested in your study.